# Pomegranate Seed Oil as a Source of Conjugated Linolenic Acid (CLnA) Has No Effect on Atherosclerosis Development but Improves Lipid Profile and Affects the Expression of Lipid Metabolism Genes in apoE/LDLR^−/−^ Mice

**DOI:** 10.3390/ijms24021737

**Published:** 2023-01-15

**Authors:** Magdalena Franczyk-Żarów, Tomasz Tarko, Anna Drahun-Misztal, Izabela Czyzynska-Cichon, Edyta Kus, Renata B. Kostogrys

**Affiliations:** 1Department of Human Nutrition and Dietetics, Faculty of Food Technology, University of Agriculture in Krakow, A. Mickiewicz Av. 21, 31-120 Kraków, Poland; 2Department of Fermentation Technology and Microbiology, Faculty of Food Technology, University of Agriculture in Krakow, A. Mickiewicz Av. 21, 31-120 Kraków, Poland; 3Jagiellonian Centre for Experimental Therapeutics (JCET), Jagiellonian University, Bobrzynskiego St. 14, 30-348 Kraków, Poland

**Keywords:** pomegranate seed oil, conjugated linolenic acid, apoE/LDLR^−/−^ mice, atherosclerosis

## Abstract

The aim of this study was to evaluate the anti-atherosclerotic effect of pomegranate seed oil as a source of conjugated linolenic acid (CLnA) (*cis*-9,*trans*-11,*cis*-13; punicic acid) compared to linolenic acid (LnA) and conjugated linoleic acid (CLA) (*cis*-9,*trans*-11) in apoE/LDLR^−/−^ mice. In the LONG experiment, 10-week old mice were fed for the 18 weeks. In the SHORT experiment, 18-week old mice were fed for the 10 weeks. Diets were supplied with seed oils equivalent to an amount of 0.5% of studied fatty acids. In the SHORT experiment, plasma TCh and LDL+VLDL cholesterol levels were significantly decreased in animals fed CLnA and CLA compared to the Control. The expression of *PPARα* in liver was four-fold increased in CLnA group in the SHORT experiment, and as a consequence the expression of its target gene *ACO* was three-fold increased, whereas the liver’s expression of *SREBP-1* and *FAS* were decreased in CLnA mice only in the LONG experiment. Punicic acid and CLA isomers were determined in the adipose tissue and liver in animals receiving pomegranate seed oil. In both experiments, there were no effects on the area of atherosclerotic plaque in aortic roots. However, in the SHORT experiment, the area of atherosclerosis in the entire aorta in the CLA group compared to CLnA and LnA was significantly decreased. In conclusion, CLnA improved the lipid profile and affected the lipid metabolism gene expression, but did not have the impact on the development of atherosclerotic plaque in apoE/LDLR^−/−^ mice.

## 1. Introduction

Conjugated fatty acids refer to a group of positional and geometric isomers of polyunsaturated fatty acids (PUFA) with conjugated double bonds, where double bonds are separated by a single C-C bond, but not by a methylene group (-CH2-). The most studied are conjugated linoleic acids (CLA) of animal origin, with two conjugated double bonds, and conjugated linolenic acids (CLnA) of vegetable origin, with three conjugated double bonds. Recently, not only for CLA but also for conjugated linolenic acid, several unique biological effects have been found. One of the highly unsaturated forms of conjugated fatty acids of linolenic acid (LnA, *cis*-9,*cis*-12,*cis*-15 C18:3 n-3) is CLnA. Punicic acid (*cis-*9,*trans*-11,*cis*-13) was found at up to 83% in pomegranate seed oil. Seed oils of certain plants include conjugated triene fatty acids such as α-eleostearic acid (α-ESA, *cis*-9,*trans*-11*,trans*-13 C18:3) and calendic acid (*trans*-8,*trans*-10,*cis*-12 C18:3) at levels of 30–80%. There is an increasing amount of specific scientific literature concerning the biological effects and properties of CLnA. In mammals, there have been reported that some isomers of -9,-11,-13 CLnA could improve the immune response [1], markers of obesity [2,3] or have an anti-carcinogenic effect [4,5,6]. In vitro studies showed among others that punicic acid reduces apolipoprotein B100 secretion and triacylglycerol synthesis in HepG2 cells [7] and inhibits breast cancer cell proliferation [8]. These properties are closely related to those of CLA, because there have been found that -9,-11,-13 CLnA are converted into *cis*-9,*trans*-11 or *trans*-9,*trans*-11 CLA isomers [9,10,11,12].

While some progress has been made in characterizing some of the health-related effects of punicic acid, its underlying mechanisms of action are not fully understood. Punicic acid exerts an anti-inflammatory effect through the inhibition of tumor necrosis factor-α (TNF-α)-induced neutrophil hyperactivation and reactive oxygen species (ROS) production and authors concluded that this natural dietary compound may provide a novel alternative therapeutic strategy in inflammatory bowel disease [13]. Punicic acid acts as an efficient antioxidant in the rat model [14,15] and showed anti-inflammatory activity [16]. In terms of the inhibitory effect on NOS activity, the expression of hepatic NF-κB, plasma TNF-α and IL-6 level, punicic acid has a potent anti-inflammatory role. However, the anti-inflammatory effect of CLnA via suppressing the synthesis of proinflammatory cytokines (i.e., IL-6, TNF-α) seems to be related to *PPARγ* expression, which is supported by the findings of Changhua [17]. Moreover, naturally occurring, orally active agonists of *PPAR*, such as CLnA, have demonstrated some promising effects as dual or PAR agonists of *PPAR* [18,19]. Punicic acid ameliorates glucose tolerance and obesity-related inflammation in animal models of obesity and type 2 diabetes by acting as dual *PPARα* and γ agonists [7,20] with no adverse side effects detected in toxicological studies [21].

Thus, as an anti-inflammatory agent and efficient antioxidant, we have attempted to evaluate if punicic acid will have influence on the development of atherosclerotic plaque. Effects of dietary supplementation with pomegranate seed oil as a source of punicic acid on the development of atherosclerosis is not known; only hypolipidemic and hypocholesterolemic effects on plasma lipids have been shown in animal and human studies [22,23]. There is no data available so far as to whether CLnA fed to apoE/LDLR^−/−^ mice could have health-related effects. Therefore, the major aim of our studies was to examine potential anti-atherogenic effects of conjugated trienoic fatty acid (C18:3 *cis*-9,*trans*-11,*cis*-13) compared to non-conjugated linolenic acid and *cis*-9,*trans*-11 CLA in a mice model of atherogenesis, e.g., apolipoprotein E and low-density lipoprotein receptor double-knockout mice (apoE/LDLR^−/−^) (Figure 1), as well as to describe the potential mechanisms involved in CLnA effects on metabolism.

## 2. Results

### 2.1. Effect of CLnA on Body Weight

The initial and final body weights of animals fed experimental diets (Table 1) are shown in Table 2. Mice fed the control and experimental diets in the SHORT study had similar initial body weight. After 10 weeks of treatment the final body weight of animals did not differ between groups. In the LONG study, due to younger mice (10 weeks old), initial body weight was lower in comparison to the SHORT study, but was similar among groups. After 18 weeks of study the body weight of mice fed the LnA diet was significantly increased compared to the control (23.17 vs. 21.58 g).

### 2.2. Effect of CLnA on Liver Weight and Histology

There were no significant differences in liver weight between the control and experimental groups in the SHORT and LONG studies (Table 2). In the same line, the histological examination of the livers showed no signs of the steatosis in mice fed diets with the addition of studied fatty acids. However, representative images of livers stained with H-E of the CLA-fed group seems to present slight fat accumulation compared to the control and LnA as well as CLnA groups (Figure 2A,B).

### 2.3. Effect of CLnA on Glucose Concentration

No significant differences in final blood glucose were observed between the experimental groups after 10 or 18 weeks of studies (Table 2). However, we observed that only the glucose concentration in the LnA group after 12 weeks of the LONG experiment was significantly (*p* < 0.05) increased compared to the control and CLA groups (143.2 mg/dL vs. 119.3 mg/dL and 117.0 mg/dL, respectively).

### 2.4. Effect of CLnA on Plasma Lipid Profile and Expression of PPARα and SREBP-1 and Their Target Genes (ACO and FAS )

The effect of dietary treatments on plasma lipid concentration is shown in Table 3. The only evident effect of inserting different fatty-acid supplementation into the diet was the significantly (*p* < 0.05) decreased plasma TCh and LDL+VLDL cholesterol levels in animals fed CLnA and CLA as compared to the effect of the control diet in the SHORT study. Moreover, the LDL+VLDL cholesterol concentration was the lowest in the CLnA group (6.94 mmol/L) compared to the control (14.51 mmol/L) and non-conjugated linolenic acid (LnA: 12.14 mmol/L). At the same time, HDL and TAG levels did not differ significantly. However, the concentration of TAG tended to decrease in mice fed the CLnA diet in comparison to the control (1.92 vs. 3.76 mmol/L). Nevertheless, no changes in lipid profile were observed in the LONG experiment.

Expression of transcription factors *PPARα* (peroxisome proliferator-activated receptor α) and *SREBP-1* (sterol regulatory element binding protein-1) and their target genes *ACO* (acyl-CoA oxidase) and *FAS* (fatty acid synthase), respectively, were detected in the liver of the animals and is shown in Table 3. The results were obtained by dividing the expression of genes mentioned above by expression *β-actin* and are presented in arbitrary units. In the SHORT experiment, the *PPARα* mRNA level differed between the four groups of mice. Mice fed the diet containing pomegranate seed oil had almost four-fold higher mRNA levels of *PPARα* in the liver than the control mice. Moreover, the same mice showed three-fold higher level of *SREBP-1* mRNA compared to the control mice. The expression of their target genes, such as *ACO* and *FAS*, were about three-fold increased in mice fed the diet in the SHORT study. In the LONG study, the *SREBP-1* mRNA level as well as expression of *FAS* were decreased in CLnA mice. The same was true for animals fed the diet containing non-conjugated linolenic acid, whereas in CLA mice the increased expressions of *SREBP-1* and *FAS* as well as *PPARα* and *ACO* were observed.

### 2.5. Effect of CLnA on Fatty Acid Profile in Tissues

The fatty acid profile in the adipose tissue and liver were investigated using chromatographic analysis.

When the fatty acid profile of adipose tissue is considered, only SFA concentration was not changed in both experiments (Table 4). Additionally, a significantly decreased level of MUFA and increased PUFA were observed in the CLA group in the SHORT study. In the same line, in the LONG study the level of MUFA was significantly decreased in the CLA group. The amount of stearic acid (C18:0) was significantly decreased in groups fed CLnA and CLA in the SHORT study. The concentration of palmitoleic acid (C16:1) was significantly increased, whereas the amount of stearic acid was significantly decreased in group fed the CLA diet in the LONG study. Moreover, the level of α-linolenic acid (C18:3) was significantly increased in the LnA group in both experiments. In groups fed pomegranate seed oil and CLA supplemented diets, the *cis*-9,*trans*-11 and *trans*-10,*cis*-12 isomers were present, but the amount of these CLA isomers was significantly increased only in the CLA group in both experiments. Additionally, in mice fed pomegranate seed oil, the supplemented-diet punicic acid was detected in the SHORT as well as in the LONG experiment.

Taking into account the fatty acid profile of liver, no significant differences between the amount of SFA, MUFA and PUFA were shown in the SHORT as well as in the LONG experiment (Table 5). The amount of α-linolenic acid (C18:3) was significantly increased in mice fed flaxseed oil supplemented diets (LnA) in both experiments. Additionally, in the SHORT study, in groups where pomegranate seed oil and CLA were provided, the *cis*-9,*trans*-11 CLA isomer was present. A the same time in the LONG experiment, not only the *cis*-9,*trans*-11 CLA isomer but also *trans*-10,*cis*-12 CLA were present, but the concentration of the *cis*-9,*trans*-11 CLA isomer was significantly increased in the group fed with the pomegranate seed oil supplemented diet. Moreover, in this group punicic acid was detected.

To summarize, the fatty acid profile was affected by different fatty-acid supplementation into the animals’ diets. Although individual isomers of CLA were not supplemented into the diet for group fed with pomegranate seed oil, CLA isomers were present in liver as well as in the adipose tissue of mice; what is more, *trans*-10,*cis*-12 was incorporated into the tissue lipids in a significantly lower proportion than *cis*-9,*trans*-11. These findings confirm that CLnA is converted into CLA isomers in model organisms.

### 2.6. Effect of CLnA on the Development of Atherosclerosis

Quantification of atherosclerotic plaque as measured in the entire aorta (*en face* method) stained using Sudan IV, showed that the only significant (*p* < 0.05) difference is in CLA-fed animals in the SHORT study. In this group, the smallest development of lesion was observed (Figure 3A,B) compared to the LnA and CLnA groups (12.83% vs. 19.47 and 20.57%, respectively). However, in the evaluated groups of the LONG experiment, a significant difference in area of atherosclerotic plaque was not observed (Figure 4A,B). Also area of the plaques as measured in aortic roots (cross-section method) did not differ significantly between experimental groups in the SHORT (Figure 5A,B) as well as in the LONG experiment (Figure 6A,B). However, in the LONG study, the area of atherosclerotic lesions tended to decrease in the CLA group (1,266,174 µm^2^), whereas in LnA-fed animals it tended to increase (1,702,467 µm^2^) compared to the control (1,347,660 µm^2^). Additionally, the diets containing studied fatty acids had no effect on macrophage accumulation in atherosclerotic plaques as determined by immunohistochemical staining for a specific macrophage marker (CD68), as well as on the area covered by SMA-positive cells in aortic plaques.

## 3. Discussion

Many studies indicated that CLnA isomers have some pro-health effects. In our experiment, the effect of pomegranate seed oil in an animal model of atherosclerosisapoE/LDLR^−/−^ mice was evaluated. Whereas, previous investigations focused mainly on the hypocholesterolemic effects of plasma lipids, we aimed to assess the development of atherosclerotic plaque, as the endpoint.

### 3.1. Effect of CLnA on Body Weight

In the SHORT study, no effects of experimental treatments on body weight of mice were evident. Only in the LONG study, in animals fed the diet containing non-conjugated linolenic acid, was an increase in final body weight observed. The lack of influence of punicic acid on animals’ body weight was observed in some studies [1,11,24]. Similarly, other authors did not observe the effect of different CLnAs isomers on body weight in model animals [18,25,26]. However, a high-fat diet (HFD) supplemented with 1% pomegranate seed oil for 12 weeks significantly decreased the body weight of mice without affecting the food intake or energy expenditure [27].

### 3.2. Effect of CLA Isomers on Liver Weight and Histology

In the current study, no effect on the liver weight of animals was observed; these results are consistent with previously reported studies [1,6,7,11]. Nevertheless, some reports indicated that a mixture of CLnA induce hepatic lipid accumulation in rats and that the relative liver weight was significantly heavier in rats fed CLnA [28]. In contrast, the liver weight was significantly decreased in animals after CLnA-isomers feeding [24,29,30]. Hypoplasia induced in the liver due to the antiproliferative effect on hepatocytes, together with the tendency toward increased transaminase levels, should be considered before proposing CLnA isomers as a functional ingredient [24]. Moreover, in some studies increased liver TAG concentration was observed [28].

In our study, the histological examination of the livers showed no signs of steatosis. However, representative images of the liver-stained H-E of CLA fed group seems to present slight fat accumulation compared to the control and LnA as well as CLnA groups. The increased liver mass produced by CLA may be explained by the accumulation of lipids derived from the mobilization of delipidated adipose tissue stores [31]. Raffaele et al. [32] showed that the supplementation of obese mice with pomegranate seed oil decreased hepatic steatosis and fibrosis. Additionally, the lipid droplet diameter was decreased when compared with the liver of mice fed a high fat diet. Authors concluded that pomegranate seed oil has the potential therapeutic effect in the prevention of obesity-derived non-alcoholic fatty liver disease as well as other metabolic disorders [32].

### 3.3. Effect of CLA Isomers on Glucose Concentration

In the current project, no significant differences in final blood glucose were observed between the experimental groups after 10 or 18 weeks of studies. Our results are consistent with previous reports [24,25,27]. However, pomegranate seed oil has been shown to decrease plasma glucose. Nekooeian et al. [33] showed that diabetic rats treated with pomegranate seed oil had significantly higher levels of serum insulin, but there were no statistically significant differences in terms of blood glucose between them and the diabetic control group. Saha and Ghosh [34] observed that administration of CLnA isomers significantly lowered the blood sugar level of STZ-induced diabetic rats as compared to the diabetic control group.

### 3.4. Effect of CLA Isomers on Plasma Lipid Profile and Expression of PPARα and SREBP-1 and Their Target Genes (ACO and FAS )

Among the markers of biological response, the serum lipid profile of the experimental mice was evaluated in detail. However, our studies produced equivocal results. The effect was evident only in the SHORT study, whereas in the LONG study all parameters of the lipid profile were unchanged. In the mice fed with CLnA isomers in the SHORT study, total plasma cholesterol as well as LDL+VLDL cholesterol were significantly decreased. Moreover, the TAG concentrations tended to decrease as compared to the effect of the control diet. Thus, in our study we confirmed hypolipidemic and hypocholesterolemic activities of punicic acid in apoE/LDLR^−/−^ mice during 10 weeks of experiment. The cholesterol-lowering effect is probably only short-term and therefore does not affect the long-term reduction of atherosclerotic plaque. It is supposed that punicic acid is more efficient in reducing TCh content than other CLnA isomers. This can be attributed to the fact that punicic acid has 66% *cis* configuration, so it might have acted like essential fatty acids such as linoleic or n-6 linolenic acid, which normally reduce the cholesterol level in plasma and other tissues [23].

The results of previous research of CLnA isomers are inconclusive. Some authors observed decreased parameters of lipid profile (TCh, LDL, TAG) [18,25,35], whereas others did not report any differences [29,36]. Additionally, hypolipidemic and hypocholesterolemic activities of CLnA isomers present in bitter gourd and snake gourd seed oils was evaluated [23]. The authors reported that CLnA isomers normalized cholesterol, LDL-cholesterol, HDL-cholesterol and triacylogliceride contents in rat-plasma-treated sodium arsenite. The results of some studies evaluating the effect of punicic acid on the lipid profile indicate no changes in the studied parameters [7,24,30,33]. Also, Koba et al. [37] and Yuan et al. [11] did not find any significant effect of punicic acid on lipids but the liver TAG levels were significantly decreased when compared to the control group. On the other hand, Koba et al. [28] showed that feeding 1% of dietary CLnA significantly increased serum and liver TAG in rats. Similarly, Yamasaki et al. [1] reported significant increases in serum TAG and phospholipid levels but not in TCh in mice fed 0.12 and 1.2% punicic acid.

Study conducted on hyperlipidaemic patients showed that concentration of TAG and the TAG:HDL cholesterol ratio were significantly decreased after 4 weeks of 400 mg pomegranate seed oil treatment (one capsule twice daily). This human study confirmed that CLnA had favorable effects on lipid profiles [22].

The attempt to explain the mechanism of CLnA isomers effects on plasma lipids was made. The findings of results seems to indicate that *trans* or *cis* configuration of conjugated trienoic fatty acids isomers affects differently. With the advent of nutritional genomics, novel mechanisms by which dietary CLnA might modulate plasma lipids have been suggested. Conjugated fatty acids (CLnA, CLA) are considered to be ligands of nuclear transcription factors which are involved in lipid hepatic metabolism. For instance, activation of *PPARα* by CLnA may lead to several effects, e.g., the reduction of fatty acid and triacylglycerol synthesis and VLDL production. The *SREBP-1* regulates genes encoding lipogenesis and cholesterol biosynthesis enzymes. CLnA may decrease plasma cholesterol and TAG levels via this mechanism. Authors indicated that CLnA affects lipid metabolism through activation of the *PPAR*.

In vitro studies have reported that conjugated linolenic acid isomers act as agonists for *PPARs* [19,38,39], a family of nuclear receptors that regulate the transcription of a large number of genes involved in numerous metabolic pathways. Verification of if CLnA isomers serve as ligands in living organisms is unavoidable. Our results showed that expression of *PPARα* in apoE/LDLR^−/−^ mice in the SHORT study was increased by the CLnA isomer, and the effect was even stronger than for CLA. Surprisingly, there was no such an effect in the LONG study. Moreover, the expression of *PPARα* was decreased by CLnA. Conjugated linolenic acids, such as punicic acid, catalpic acid and eleostearic acid have demonstrated some promising effects as dual or PAN agonists of *PPAR* [18,19]. Punicic acid supplementation did not modify *ACO* gene expression as well as *PPARδ* [24] in the rats’ liver. Moreover, a significant increase in *PPARα* expression was shown. These results are in line with those reported by Arao et al. [7]. Bassaganya-Riera et al. [3] shown that punicic acid regulates macrophage and T-cell function through *PPARγ*- and δ-dependent mechanisms. We hypothesized that a potent anti-atherosclerotic effect of punicic acid in apoE/LDLR^−/−^ mice could occur due to anti-inflammatory efficacy, primarily by targeting *PPAR*. At the molecular level, *PPAR* represent important targets for the actions of dietary lipid.

The results of our studies are inconclusive; we showed that the expression of *SREBP-1* in apoE/LDLR^−/−^ mice in the SHORT study was increased by the CLnA isomer, and that the effect was even stronger than for CLA. In contrast, the mRNA levels of *SREBP-1* and *FAS* were decreased by punicic acid in the LONG study. The decreased expression of genes related to fatty acid synthesis also suggest that punicic acid can directly inhibit *SREBP-1* transcription in a similar manner to PUFA but only in the LONG study. Shinohara et al. [26] showed that jacardic acid fed to ICR mice decreased the expression level of *SCD-1* mRNA causing inhibition of *SCD* activity which may have anti-obesity and anti-diabetes effects. *SCD-1* and *FAS* are regulated via *SREBP-1* and PUFA modulates *SCD-1* by directly inhibiting *SREBP-1c* transcription.

### 3.5. Effect of CLA on Fatty Acid Profile in Tissues

In our experiment carried out on apoE/LDLR^−/−^ mice, we have evaluated the degree of exposure of the mice to dietary punicic acid as well as the *cis*-9,*trans*-11 CLA isomer by measuring the CLnA and CLA isomer concentrations in the adipose tissue and liver.

It was observed that experimental treatments had a significant effect on the fatty-acid profile in adipose tissue, where MUFA were decreased at the same time PUFA were increased; SFA were unchanged. These findings are in line with the results of Shinohara et al. [26] who analyzed the fatty-acid profile of different tissue of ICR mice fed jacardic acid. They reported that levels of MUFA in liver and white adipose tissue were also decreased. As we expected, in groups fed flaxseed oil the amount of non-conjugated linolenic acid was significantly increased in the adipose tissue and liver. In groups fed pomegranate seed oil and CLA-supplemented diets, the *cis*-9,*trans*-11 and *trans*-10,*cis*-12 isomers were present, but the amount of these CLA isomers was significantly increased only in th CLA group in both experiments. Additionally, in mice fed the pomegranate seed oil-supplemented diet, punicic acid was detected in both experiments. Many studies have confirmed that CLnA incorporates into the tissue lipids of model animals following dietary CLnA consumption [40]. The present study confirmed that CLnA isomers can be converted to *cis*-9,*trans*-11 CLA. It was shown that CLnAs are not only well absorbed into the body but also metabolized to CLA in vivo [9,10,11] as well as in vitro [12]. Moreover, human studies also show that feeding of punicic acid was clearly reflected by the elevated blood concentration of CLnA, which is incorporated into red blood cell membranes and human plasma [41]. Saha and Ghosh [42] reported that CLnA isomers caused the restoration of renal fatty acid altered by aresnite-induced renal oxidative stress. Moreover, the same authors showed that CLnA isomers are effective in protecting tissue lipid profiles which were altered due to oxidative stress in rats [16]. Additionally, supplementation of CLnA isomers restored erythrocyte membrane lipid peroxidation and lipoprotein oxidation in rats with induced oxidative stress [43]. In contrast, Yuan et al. [44] discovered that supplementation with punicic acid of healthy humans caused a significantly elevated concentration of 8-iso-PGF2α in urine, which indicates that punicic acid increases lipid peroxidation in humans.

### 3.6. Effect of CLnA on the Development of Atherosclerosis

The development of atherosclerosis in apoE/LDLR^−/−^ mice, which represent a unique and reliable model of atherosclerosis [45,46,47] that allows the quantification of anti-atherogenic activities of food components, was studied for the first time.

The area of atherosclerotic lesions evaluated using the *en face* method did not vary between CLnA and LnA. The only significant effect was shown for the CLA group compared to LnA and CLnA, where a decreased area of atherosclerosis was observed. The same effect of *cis*-9,*trans*-11 CLA was observed by Arbones-Mainar et al. [48]. However, in Nestel et al. [49], supplemental 0.9% CLA (*cis*-9,*trans*-11) to the diabetic apoE-deficient mouse failed to reduce the severity of aortic atherosclerosis. In our previous studies on apoE/LDLR^−/−^ mice fed CLA-enriched egg yolks, we observed no significant changes in the amount of atherosclerosis, but we reported the stabilization of the plaque introduced as lower-macrophage accumulation (CD68), and the greater area covered by SMA-positive cells in aortic plaques in CLA-enriched eggs [50]. Moreover, we have demonstrated that CLnA supplemented to margarine did not possess an anti-atherosclerotic effect [51].

Despite metabolic changes, no atherosclerotic plaque size differences between groups have been observed in the current study. In order to assess the atherosclerosis, two methods have been used (*en face* and cross-section). These quantitative methods evaluate the atherosclerotic plaque area only.

The limitation of the study was the lack of the qualitative assessment, e.g., the composition of the atherosclerotic plaque. Masson trichrome staining could be used to evaluate the collagen accumulation. Immunohistochemistry analysis would show the composition of the atherosclerotic plaque, such as the content of macrophages and SMA. It could be assessed whether the lesions were stable. The most appropriate tissue for such an assessment is the brachiocephalic artery (BCA); unfortunately this vessel was not collected.

## 4. Materials and Methods

### 4.1. Animals and Housing

Animals (apolipoprotein E and low-density lipoprotein receptor double-knockout mice apoE/LDLR^−/−^) were provided by Taconic (Ejby, Denmark). ApoE/LDLR^−/−^ mice spontaneously develop severe atherosclerotic plaques as well as hyperlipidaemia without using cholesterol-rich diets, representing unique and reliable model of atherogenesis. In the studies using the single knockout mice models (apoE or LDLR deficient mice), the high fat diet should be used. Such a factor could interfere the results of the study. Using the double knockout mice model (apoE/LDLR^−/−^) it is possible to avoid the diet modification (such as high fat diet). Female mice were used in the study. According to Man et al. [52] male animals appear to have more inflamed but smaller plaques compared to female animals.

Animals were housed in colony cages in a 22–25 °C with a 12h light/dark cycle. They were fed ad libitum.

All procedures involving animals were approved by the 1st Local Animal Ethics Commission in Krakow.

### 4.2. Diets and Feeding

8–10 weeks old (no signs of atherosclerosis) and 16–18 weeks old (pre-established atherosclerosis) female mice were used in this study. Up to the age of 10 or 18 weeks, the mice were fed a commercial, cholesterol-free, pelleted diet (Sniff M-Z Spezialdiäten GmbH; Soest, Germany).

The nutritional experiment was conducted in two steps. Based on our previous studies [53] it was observed that atherosclerotic plaque in apoE/LDLR^−/−^ mice develops much faster comparing to single knockout. Therefore, we decided that the experimental design will be divided into two parts (SHORT—regression and LONG—progression).

Mice were randomly assigned to four experimental groups (*n* = 10 in each group) at the age of 10 or 18 weeks and fed AIN-93G-based diets [54] for the next 18 weeks (LONG) or 10 weeks (SHORT), respectively.

The experimental design is demonstrated in Figure 1 and the composition of modified diets fed to model animals is shown in Table 1.

The following experimental diets were used: I—control (AIN-93G), II—flaxseed oil, LnA (as a source of linolenic acid), III—pomegranate seed oil, CLnA (as a source of punicic acid), IV—CLA (*cis*-9,*trans*-11). The amount of 0.5% of studied fatty acids was supplied with seed oils to the diets.

At the end of the nutritional experiment the mice were injected intraperitoneally (1000 IU of heparin, Sanofi-Synthelabo; Paris, France) and after 10 min anesthetized (40 mg/kg of sodium thiopental, Biochemie; Vienna, Austria) and sacrificed by cervical translocation.

### 4.3. Blood Sampling, Plasma Lipid Profile and Glucose Analyses

To obtain plasma samples, blood was taken from the left ventricle of the heart and centrifuged (4000× *g*, 10 min). Until further analysis the samples were stored in −80 °C. Lipid profile was analyzed using commercially available kits for: total cholesterol (TCh; Liquick Cor-Chol 60 No 2-204; Cormay, Lublin, Poland), HDL-cholesterol (HDL; Olympus Diagnostica GmbH No OSR 6287; Hamburg, Germany) and triacylglycerols (TAG; Liquick Cor-TG 30 No 2-262; Cormay, Lublin, Poland). The LDL+VLDL-cholesterol were calculated as the difference between TCh and HDL-cholesterol. Blood glucose concentrations were measured with an Accu-Chek Active strips (Roche Diagnostics GmbH, Mannheim, Germany).

### 4.4. Fatty Acid Profile of Tissues

The abdominal adipose tissue and part of liver were removed from the mice and snap frozen −80 °C. The lipid profile of adipose tissue and liver were analyzed using GC-MS analyzer (Shimadzu GC-MS, Model QP 5050A) [55].

### 4.5. Quantification of Atherosclerosis in Descending Aortas (en Face Analysis)

In anesthetized mice, the thorax was longitudinally opened, the right atrium was incised, and the heart was perfused with phosphate-buffered saline (PBS, pH 7.4) through the apex of the left ventricle. The entire aorta was dissected and fixed in 4% paraformaldehyde, opened longitudinally, pinned and stained with Sudan IV (Sigma-Aldrich, St. Louis, MO, USA) [50]. The aortic lesion area was measured using LSM Image Browser 3 software (Zeiss, Jena, Germany).

### 4.6. Quantitation of Atherosclerosis in Aortic Roots (Cross-Section Analysis)

Heart and the ascending aorta were dissected. The detailed method was described previously [51]. Briefly, the heart and ascending aorta were embedded in OCT compound (CellPath, Oxford, UK), frozen and cut in 10 μm-thick serial cryosections. Eight sections were collected at 100 μm intervals starting at a 100 μm distance from the appearance of the aortic valves. Sections were fixed and stained with oil-red-O (Sigma-Aldrich, St. Louis, MO, USA) and examined under an Olympus BX50 (Olympus, Tokyo, Japan) microscope. Images of the aorta were recorded using an Olympus Camedia 5050 digital camera and stored as TIFF files of resolution 1024 × 768 pixels. The area of the lesion was measured using LSM Image Browser 3 software (Zeiss, Jena, Germany). For each animal, a mean lesion area was calculated from eight sections, reflecting the cross-section area covered by atherosclerosis.

### 4.7. Liver Histological Examination

Liver was removed, fixed in 4% buffered paraformaldehyde (pH 7.4) and dehydrated in alcohol, embedded in paraffin and sectioned serially (7 µm). Then the sections were stained with hematoxylin and eosin [53].

### 4.8. Real-Time qRT-PCR

The detailed method was described previously [24]. Briefly, total RNA kit (A&A Biotechnology, Gdańsk, Poland) was used according to the manufacturer’s protocol. RNA quantity was measured using NanoDrop (NanoDrop Technologies, Wilmington, DE, USA) and BioAnalyzer (Agilent, Santa Clara, CA, USA).

Expression of the selected genes was checked using qRT-PCR with the following primers:

*PPAR*-F 5′-GAGGGTTGAGCTCAGTCAGG-3′

*PPAR*-R 5′-GGTCACCTACGAGTGGCATT-3′,

*SREBP*-F 5′-CACTCAGCAGCCACCATCTA-3′

*SREBP*-R 5′-GCTGTCAGCAGCAGTGAGTC-3′

*ACO*-F 5′-GCT GGC CGT GTC CAT AGC-3′

*ACO*-R 5′-TTA TCC GTG GGT CCA AAC TGA-3′,

*FAS*-F 5′-TTG CTG GCA CTA CAG AAT GC-3′

*FAS*-R 5′-AAC AGC CTC AGA GCG ACA AT-3′,

*β-actin*-F: 5′-AGCCATGTACGTAGCCATCCA-3′

*β-actin*-R: 5′-TCTCAGCTGTGGTGGTGAAG-3′ (primer sequences designed in Primer3 software, based on sequence BC102589 from the GenBank). We used a LightCycler FastStart DNA Master SYBR Green I kit (Roche Diagnostics, Germany) according to the following procedure: Mg^2+^ added at a final concentration of 3 mM; preincubation step at 95 °C for 10 min; amplification step (40 cycles) including denaturation at 95 °C for 10 s, annealing at 58 °C for 10 s, extension at 72 °C for 10 s; melting curve including denaturation at 95 °C for 0 s, annealing at 65 °C for 15 s, continuous melting at 95 °C for 0 s (slope = 0.1 °C s^−1^); and cooling step at 40 °C for 30 s. For *β-actin*, annealing was at 58 °C. Results are presented as the ratio of genes to *β-actin* expression. The housekeeping gene *β-actin* was used for normalization.

### 4.9. Statistical Analysis

The one-way analysis of variance ANOVA calculated using STATISTICA 8.0 package (StatSoft Inc., Tulsa OK Oklahoma, USA) was used. To compare the treatment means the Duncan’s multiple range test was applied. A significant difference between treatment means was considered for *p* values < 0.05.

## 5. Conclusions

Since the CLnAs are present in some plant seeds in large amounts and could be purified relatively easily, they could be alternative sources for obtaining the health effects of themselves as well of CLAs. Therefore, CLnAs, which are more available and have a stronger physiological effect than CLA, are expected to be effective as a physiologically active lipid. These natural dietary compounds may provide a novel alternative therapeutic strategy in many diseases such as inflammatory diseases and cancer. Although we failed to prove an anti-atherosclerotic effect assessed directly on the atheroma area, we confirmed hypocholesterolemic properties of serum lipids. Moreover, we have attempted to understand the molecular mechanism of action of CLnA by determining gene expression. More studies are warranted to evaluate if natural resources containing CLnAs could be a potential dietary source of CLA and if they are expected to be useful as a functional food and nutraceuticals.

## Figures and Tables

**Figure 1 ijms-24-01737-f001:**
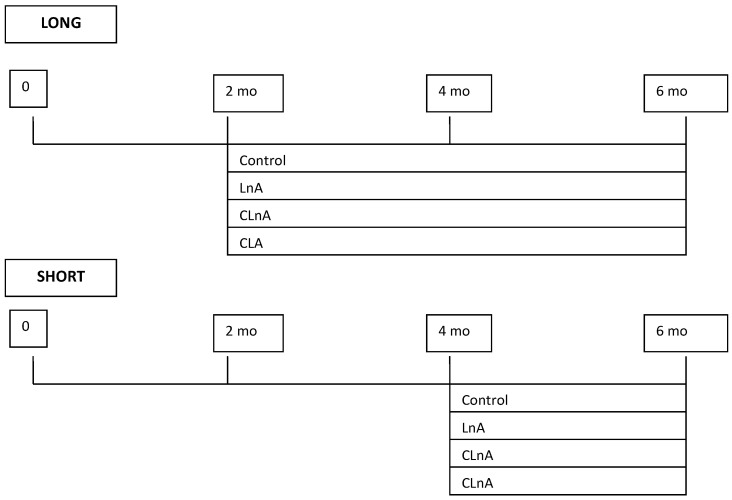
Experimental design.

**Figure 2 ijms-24-01737-f002:**
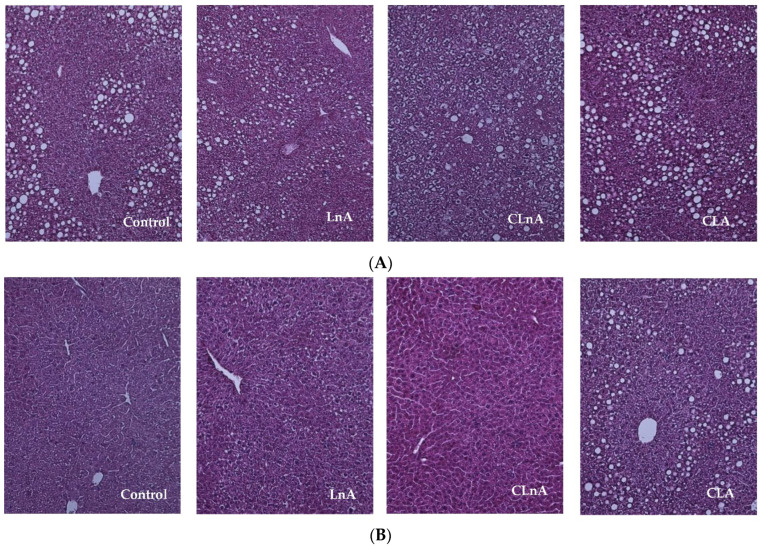
(**A**) Representative images of liver (stained hematoxylin and eosin) in apoE/LDLR^−/−^ mice (*n* = 10) fed experimental diets in SHORT. (**B**) Representative images of liver (stained hematoxylin and eosin) in apoE/LDLR^−/−^ mice (*n* = 10) fed experimental diets in LONG. Magnification of 200x.

**Figure 3 ijms-24-01737-f003:**
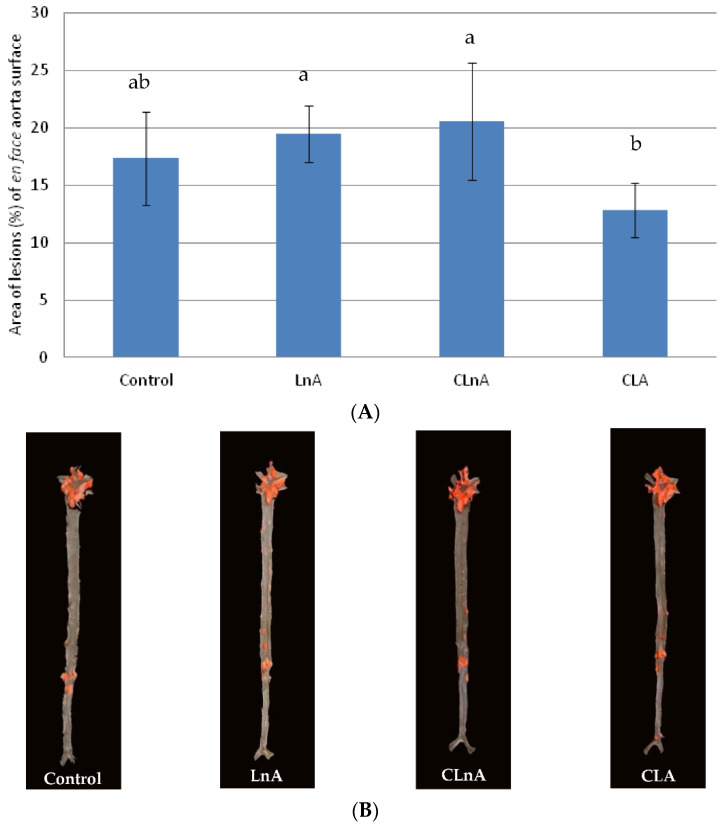
(**A**) Quantification (% area) of atherosclerotic plaque in entire aorta (*en face*) in apoE/LDLR^−/−^ mice (*n* = 10) fed experimental diets in SHORT. Values are means with their standard errors depicted by vertical bars. The means marked with letters a and b are significantly different (*p* < 0.05). (**B**) Representative images of *en face* aorta showing atherosclerotic plaque in apoE/LDLR^−/−^ mice (*n* = 10) fed experimental diets in SHORT.

**Figure 4 ijms-24-01737-f004:**
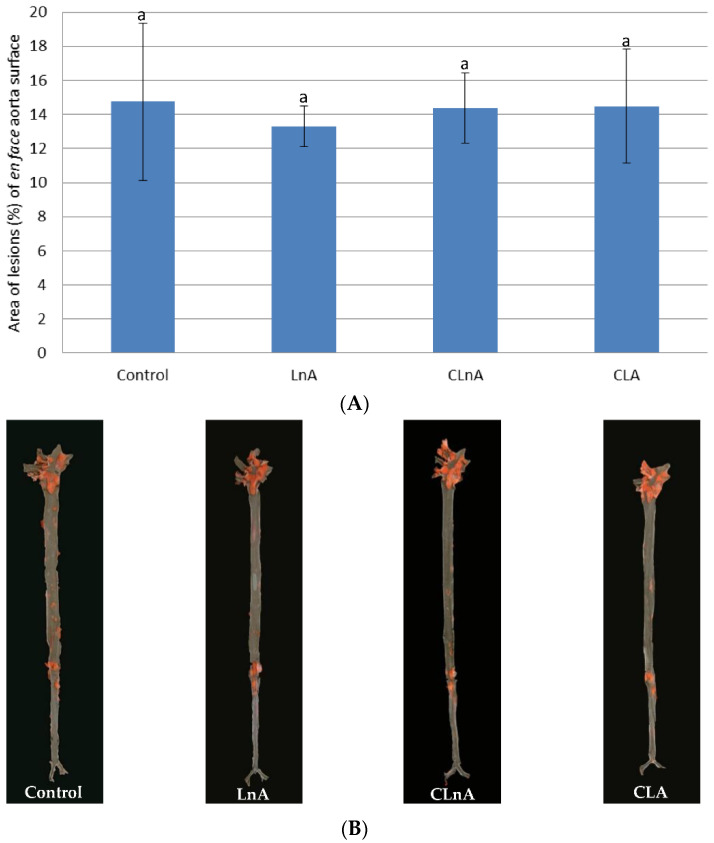
(**A**) Quantification (% area) of atherosclerotic plaque in entire aorta (*en face*) in apoE/LDLR^−/−^ mice (*n* = 10) fed experimental diets in LONG. Values are means with their standard errors depicted by vertical bars. The means marked with letters a are not significantly different (*p* < 0.05). (**B**) Representative images of *en face* aorta showing atherosclerotic plaque in apoE/LDLR^−/−^ mice (*n* = 10) fed experimental diets in LONG.

**Figure 5 ijms-24-01737-f005:**
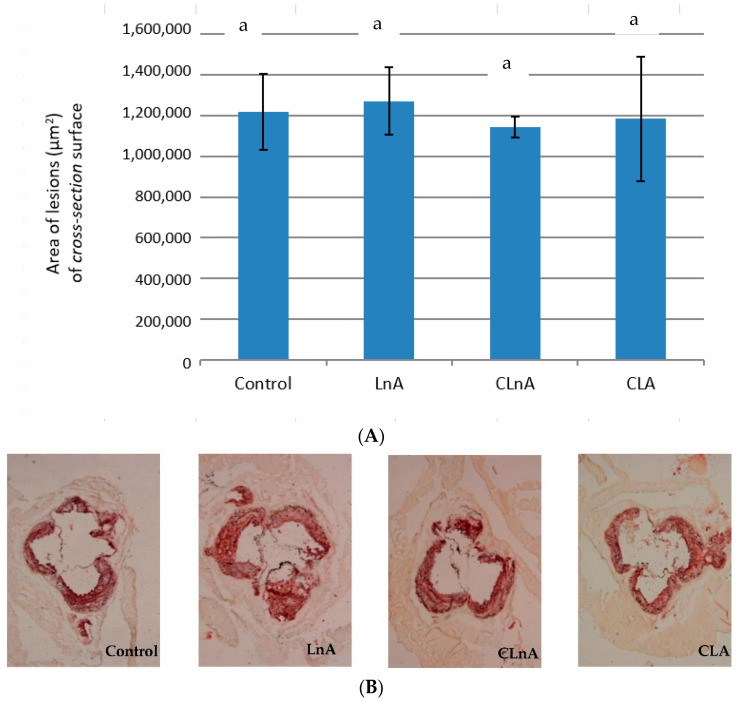
(**A**) Area (µm^2^) of atherosclerosis of aortic roots (*cross-section*) in apoE/LDLR^−/−^ mice (*n* = 10) fed experimental diets in SHORT. Values are means with their standard errors depicted by vertical bars. The means marked with letters a are not significantly different (*p* < 0.05). (**B**) Representative images of cross-section of aortic roots (stained ORO) showing aortic plaque in apoE/LDLR^−/−^ mice (*n* = 10) fed experimental diets in SHORT. Magnification of 50×.

**Figure 6 ijms-24-01737-f006:**
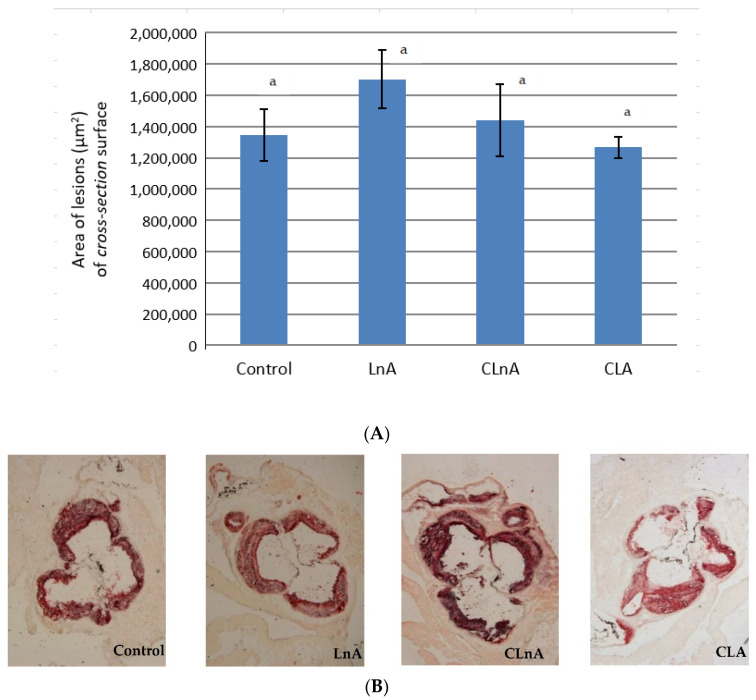
(**A**) Area (µm^2^) of atherosclerosis of aortic roots (cross-section) in apoE/LDLR^−/−^ mice (*n* = 10) fed experimental diets in LONG. Values are means with their standard errors depicted by vertical bars. The means marked with letters a are not significantly different (*p* < 0.05). (**B**) Representative images of cross-section of aortic roots (stained ORO) showing aortic plaque in apoE/LDLR^−/−^ mice (*n* = 10) fed experimental diets in LONG. Magnification of 50×.

**Table 1 ijms-24-01737-t001:** Composition of experimental diets (g/kg).

	Control	LnA	CLnA	CLA
Corn starch	532.486	532.486	532.486	532.486
Caseine	200	200	200	200
Sucrose	100	100	100	100
Soybean oil	70	59.75	61.3	63.34
Celulose powder	50	50	50	50
Mineral mixture ^a^	35	35	35	35
Vitamin mixture ^b^	10	10	10	10
Choline	2.5	2.5	2.5	2.5
tert-butylhydrochinon	0.014	0.014	0.014	0.014
Flaxseed oil ^c^	-	10.25	-	-
Pomegranate seed oil ^d^	-	-	8.7	-
*cis*-9,*trans*-11 CLA ^e^	-	-	-	6.66

^a^ AIN-93G mineral mixture. ^b^ AIN-93G vitamin mixture. ^c^ Flaxseed oil (Earthoil Plantations Ltd. Bury St Edmunds, UK) contained 48.8 g LnA/100 g. ^d^ Pomegranate seed oil (Earthoil Plantations Ltd. Bury St Edmunds UK) contained 57.5 g CLnA/100 g. ^e^ CLA isomers (Larodan Fine Chemicals) *cis*-9,*trans*-11 CLA methyl ester, purity 75%.

**Table 2 ijms-24-01737-t002:** Body weight (g), glucose (mg/dL), total cholesterol (TCh) (mmol/L), triacylglycerols (TAG) (mmol/L) level in plasma in apoE/LDLR^−/−^ mice (*n* = 10) fed experimental diets. The means bearing different letters were significantly different (*p* < 0.05). Small letters indicate a separate comparison between groups in SHORT experiment. Capital letters indicate a separate comparison between groups in LONG experiment.

	SHORT				LONG			
	Control	LnA	CLnA	CLA	Control	LnA	CLnA	CLA
Initial body weight (g)	23.06 ± 1.02	21.77 ± 1.68	22.91 ± 2.04	22.17 ± 1.62	17.82 ± 1.45	18.97 ± 1.06	18.13 ± 1.83	17.46 ± 1.51
Final body weight (g)	23.25 ± 0.74 ^a^	22.13 ± 2.32 ^a^	22.97 ± 2.44 ^a^	21.94 ± 0.97 ^a^	21.58 ± 1.56 ^A^	23.17 ± 1.25 ^B^	21.97 ± 1.20 ^AB^	21.24 ± 2.09 ^A^
Liver weight (g/100g BW)	4.45 ± 0.35 ^a^	4.19 ± 0.45 ^a^	4.70 ± 0.58 ^a^	4.58 ± 0.52 ^a^	4.28 ± 0.54 ^A^	4.36 ± 0.39 ^A^	4.61 ± 0.61 ^A^	4.37 ± 0.46 ^A^
Final glucose (mg/dL)	135.2 ± 12.5 ^a^	144.1 ± 25.2 ^a^	140.2 ± 19.3 ^a^	122.8 ± 15.4 ^a^	120.4 ± 11.6 ^A^	123.6 ± 14.8 ^A^	133.0 ± 13.7 ^A^	123.2 ± 9.4 ^A^
TCh (mmol/L)	28.59 ± 5.06 ^b^	23.67 ± 2.45 ^ab^	19.93 ± 5.29 ^a^	21.84 ± 0.58 ^a^	20.60 ± 1.24 ^A^	20.78 ± 2.49 ^A^	21.05 ± 2.45 ^A^	21.82 ± 1.69 ^A^
LDL + VLDL (mmol/L)	14.51 ± 1.97 ^c^	12.14 ± 2.19 ^bc^	6.94 ± 0.73 ^a^	10.57 ± 1.10 ^ab^	8.31 ± 0.76 ^A^	8.56 ± 1.51 ^A^	8.90 ± 1.50 ^A^	9.20 ± 1.27 ^A^
HDL (mmol/L)	0.94 ± 0.22 ^a^	0.73 ± 0.15 ^a^	0.78 ± 0.13 ^a^	0.65 ± 0.03 ^a^	0.74 ± 0.13 ^A^	0.78 ± 0.15 ^A^	0.78 ± 0.19 ^A^	0.80 ± 0.16 ^A^
TAG (mmol/L)	3.76 ± 1.59 ^a^	2.27 ± 0.30 ^a^	1.92 ± 0.09 ^a^	2.46 ± 0.80 ^a^	1.85 ± 0.52 ^A^	2.04 ± 0.81 ^A^	1.96 ± 0.74 ^A^	2.12 ± 0.95 ^A^

**Table 3 ijms-24-01737-t003:** Genes expression (peroxisome proliferator-activated receptor; *PPARα* sterol regulatory element binding proteins; SREBP-1, acyl-CoA oxidase; *ACO* and fatty acid synthase; *FAS* ) in apoE/LDLR^−/−^ mice (*n* = 10) fed experimental diets.

	SHORT				LONG			
	Control	LnA	CLnA	CLA	Control	LnA	CLnA	CLA
*PPARα* (a.u.)	1.00	1.32	3.77	1.94	1.00	1.97	0.93	2.58
*SREBP-1* (a.u.)	1.00	1.49	3.25	1.23	1.00	0.73	0.78	3.14
*ACO* (a.u.)	1.00	0.72	2.83	0.66	1.00	1.01	1.11	1.71
*FAS* (a.u.)	1.00	0.72	2.83	0.66	1.00	0.69	0.56	1.16

**Table 4 ijms-24-01737-t004:** Fatty acid composition in adipose tissue of apoE/LDLR^−/−^ mice fed experimental diets. The means bearing different letters were significantly different (*p* < 0.05). Small letters indicate a separate comparison between groups in SHORT experiment. Capital letters indicate a separate comparison between groups in LONG experiment.

Fatty Acid	SHORT				LONG			
	Control	LnA	CLnA	CLA	Control	LnA	CLnA	CLA
C 14:0	1.37 ± 0.24 ^a^	1.44 ± 0.20 ^a^	1.34 ± 0.27 ^a^	1.30 ± 0.30 ^a^	1.15 ± 0.09 ^A^	1.29 ± 0.15 ^A^	1.30 ± 0.11 ^AB^	1.47 ± 0.18 ^B^
C 16:0	16.93 ± 1.32 ^a^	17.33 ± 1.29 ^a^	18.31 ± 1.26 ^a^	17.50 ± 1.17 ^a^	17.01 ± 1.04 ^AB^	15.71 ± 0.86 ^A^	16.74 ± 0.69 ^A^	18.74 ± 2.43 ^B^
C 16:1	4.16 ± 0.61 ^a^	4.43 ± 0.80 ^ab^	4.63 ± 0.61 ^ab^	5.21 ± 0.78 ^ab^	4.13 ± 0.84 ^A^	3.94 ± 0.54 ^A^	4.64 ± 0.90 ^A^	5.62 ± 0.27 ^B^
C 18:0	4.42 ± 0.42 ^b^	4.39 ± 0.37 ^b^	3.44 ± 0.57 ^a^	2.88 ± 0.65 ^a^	3.37 ± 0.58 ^A^	4.12 ± 0.51 ^A^	3.51 ± 0.62 ^A^	2.48 ± 0.79 ^B^
C 18:1	35.45 ± 1.01 ^b^	34.73 ± 1.34 ^ab^	33.41 ± 1.04 ^a^	33.44 ± 1.61 ^a^	37.23 ± 0.95 ^BC^	38.09 ± 1.03 ^C^	35.77 ± 1.01 ^AB^	35.57 ± 1.69 ^A^
C 18:2	34.09 ± 1.92 ^b^	31.25 ± 1.56 ^a^	32.52 ± 2.17 ^ab^	32.78 ± 1.95 ^ab^	33.79 ± 1.46 ^B^	30.77 ± 0.89 ^A^	30.74 ± 1.01 ^A^	30.47 ± 2.14 ^A^
C 20:0	0.30 ± 0.05 ^a^	0.28 ± 0.07 ^a^	0.24 ± 0.06 ^a^	0.22 ± 0.08 ^a^	0.23 ± 0.04 ^A^	0.30 ± 0.05 ^C^	0.23 ± 0.06 ^A^	0.17 ± 0.03 ^B^
C 20:1	1.12 ± 0.19 ^b^	0.88 ± 0.17 ^a^	0.82 ± 0.09 ^a^	0.83 ± 0.11 ^a^	0.83 ± 0.15 ^A^	1.03 ± 0.10 ^C^	0.85 ± 0.18 ^A^	0.62 ± 0.08 ^B^
C 18:3	2.17 ± 0.32 ^b^	5.28 ± 0.48 ^c^	1.64 ± 0.11 ^a^	1.89 ± 0.22 ^ab^	1.85 ± 0.19 ^A^	4.77 ± 0.74 ^B^	1.79 ± 0.33 ^A^	1.37 ± 0.17 ^A^
CLA *cis*-9,*trans*-11			1.76 ± 0.16 ^a^	3.33 ± 0.30 ^b^	0.37 ± 0.14 ^A^		2.24 ± 0.24 ^B^	2.89 ± 0.26 ^C^
CLA *trans*-10,*cis*-12			0.52 ± 0.11 ^a^	0.65 ± 0.10 ^b^	0.16 ± 0.04 ^B^		0.69 ± 0.16 ^A^	0.59 ± 0.09 ^A^
CLnA *cis*-9,*trans*-11,*cis*-13			1.37 ± 0.27				1.51 ± 0.47	
SFA	23.01 ± 1.70 ^a^	23.43 ± 1.50 ^a^	23.33 ± 1.73 ^a^	21.87 ± 1.85 ^a^	21.76 ± 1.39 ^A^	21.41 ± 1.25 ^A^	21.77 ± 0.89 ^A^	22.86 ± 3.50 ^A^
MUFA	40.72 ± 2.69 ^a^	40.04 ± 2.77 ^ab^	38.85 ± 2.77 ^ab^	39.48 ± 2.17 ^b^	42.19 ± 2.83 ^A^	43.06 ± 2.10 ^AB^	41.25 ± 1.65 ^AB^	41.81 ± 2.03 ^B^
PUFA	36.27 ± 1.97 ^a^	36.54 ± 1.58 ^ab^	37.82 ± 2.13 ^ab^	38.65 ± 1.82 ^b^	36.16 ± 1.41 ^A^	35.53 ± 1.49 ^A^	36.97 ± 0.93 ^A^	35.33 ± 2.43 ^A^
16:1/16:0	0.25 ± 0.02 ^a^	0.26 ± 0.05 ^ab^	0.25 ± 0.03 ^ab^	0.30 ± 0.04 ^b^	0.24 ± 0.05 ^A^	0.25 ± 0.04 ^AB^	0.28 ± 0.05 ^AB^	0.30 ± 0.04 ^B^
18:1/18:0	8.03 ± 0.77 ^a^	7.91 ± 0.72 ^ab^	9.72 ± 1.41 ^b^	11.60 ± 2.52 ^c^	11.05 ± 1.80 ^A^	9.25 ± 1.07 ^A^	10.20 ± 1.71 ^A^	14.34 ± 4.21 ^B^
16:0/18:2	0.50 ± 0.06 ^a^	0.55 ± 0.06 ^a^	0.56 ± 0.06 ^a^	0.53 ± 0.06 ^a^	0.50 ± 0.04 ^A^	0.51 ± 0.04 ^A^	0.54 ± 0.03 ^A^	0.62 ± 0.13 ^B^

**Table 5 ijms-24-01737-t005:** Fatty-acid composition in liver of apoE/LDLR^−/−^ mice fed experimental diets. The means bearing different letters were significantly different (*p* < 0.05). Small letters indicate a separate comparison between groups in SHORT experiment. Capital letters indicate a separate comparison between groups in LONG experiment.

Fatty Acid	SHORT				LONG			
	Control	LnA	CLnA	CLA	Control	LnA	CLnA	CLA
C 14:0	0.56 ± 0.11 ^ab^	0.48 ± 0.05 ^ab^	0.38 ± 0.07 ^a^	0.61 ± 0.09 ^b^	0.48 ± 0.05 ^A^	0.45 ± 0.07 ^A^	0.52 ± 0.13 ^A^	0.52 ± 0.17 ^A^
C 16:0	25.80 ± 0.48 ^ab^	25.83 ± 1.74 ^ab^	24.67 ± 1.11 ^a^	28.00 ± 0.41 ^b^	24.51 ± 2.74 ^A^	24.41 ± 0.88 ^A^	22.95 ± 2.13 ^A^	24.33 ± 4.32 ^A^
C 16:1	2.21 ± 0.21 ^bc^	1.74 ± 0.18 ^ab^	1.25 ± 0.24 ^a^	2.39 ± 0.21 ^c^	1.61 ± 0.23 ^A^	1.63 ± 0.58 ^A^	2.13 ± 0.56 ^A^	2.54 ± 0.48 ^A^
C 18:0	4.23 ± 0.51 ^a^	4.26 ± 0.74 ^a^	3.84 ± 0.52 ^a^	3.47 ± 0.27 ^a^	4.31 ± 1.06 ^A^	3.47 ± 0.87 ^A^	5.15 ± 0.49 ^A^	3.97 ± 0.88 ^A^
C 18:1	30.64 ± 0.30 ^a^	31.63 ± 3.44 ^a^	35.72 ± 2.42 ^a^	30.46 ± 0.85 ^a^	34.52 ± 3.53 ^A^	31.44 ± 2.15 ^A^	33.10 ± 1.81 ^A^	32.34 ± 1.23 ^A^
C 18:2	35.13 ± 1.17 ^b^	32.97 ± 0.65 ^ab^	31.63 ± 1.66 ^a^	32.34 ± 0.77 ^ab^	32.92 ± 2.23 ^A^	34.60 ± 2.26 ^A^	31.72 ± 1.96 ^A^	33.12 ± 2.68 ^A^
C 18:3	1.44 ± 0.03 ^a^	3.08 ± 0.59 ^b^	1.18 ± 0.23 ^a^	1.27 ± 0.04 ^a^	1.66 ± 0.47 ^A^	4.01 ± 0.52 ^B^	1.82 ± 0.17 ^A^	1.79 ± 0.32 ^A^
CLA *cis*-9,*trans*-11			1.34 ± 0.30 ^a^	1.31 ± 0.03 ^a^			1.62 ± 0.18 ^B^	1.20 ± 0.22 ^A^
CLA *trans*-10,*cis*-12							0.20 ± 0.05 ^A^	0.20 ± 0.02 ^A^
CLnA *cis*-9,*trans*-11,*cis*-13							0.81 ± 0.11	
SFA	30.59 ± 0.83 ^a^	30.57 ± 2.40 ^a^	28.88 ± 0.71 ^a^	32.08 ± 0.34 ^a^	29.30 ± 2.49 ^A^	28.33 ± 1.54 ^B^	28.62 ± 2.19 ^A^	28.82 ± 4.01 ^A^
MUFA	32.84 ± 0.50 ^a^	33.37 ± 3.29 ^a^	36.97 ± 2.18 ^a^	32.85 ± 1.05 ^a^	36.13 ± 3.57 ^A^	33.06 ± 1.83 ^A^	35.22 ± 1.80 ^A^	34.88 ± 0.87 ^A^
PUFA	36.57 ± 1.14 ^a^	36.06 ± 1.24 ^a^	34.15 ± 1.73 ^a^	34.92 ± 0.79 ^a^	34.58 ± 2.47 ^A^	38.61 ± 2.34 ^A^	36.17 ± 2.24 ^A^	36.30 ± 3.20 ^A^
16:1/16:0	0.09 ± 0.01 ^b^	0.07 ± 0.00 ^a^	0.05 ± 0.01 ^a^	0.09 ± 0.01 ^b^	0.07 ± 0.01 ^A^	0.07 ± 0.02 ^A^	0.09 ± 0.02 ^AB^	0.10 ± 0.01 ^B^
18:1/18:0	7.31 ± 0.97 ^a^	7.42 ± 1.93 ^a^	9.36 ± 0.70 ^a^	8.82 ± 0.90 ^a^	8.55 ± 2.94 ^A^	9.56 ± 2.95 ^A^	6.49 ± 0.88 ^A^	8.45 ± 1.85 ^A^
16:0/18:2	0.74 ± 0.04 ^a^	0.78 ± 0.05 ^ab^	0.78 ± 0.03 ^ab^	0.87 ± 0.03 ^b^	0.75 ± 0.10 ^A^	0.71 ± 0.07 ^A^	0.73 ± 0.11 ^A^	0.75 ± 0.18 ^A^

## Data Availability

Not applicable.

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
