# Peer review of "Pomegranate Seed Oil as a Source of Conjugated Linolenic Acid (CLnA) Has No Effect on Atherosclerosis Development but Improves Lipid Profile and Affects the Expression of Lipid Metabolism Genes in apoE/LDLR^−/−^ Mice"

_ijms, 2023, doi:10.3390/ijms24021737_

Round 1

Reviewer 1 Report

The authors examine the effects of conjugated linolenic acid and linolenic acid on atherosclerosis by feeding apoE / LDLR double-deficient mice with supplemental diets, for either 10 or 18 weeks. At the time of tissue harvest, mice have the same age (and are comparable).

Although some changes in body weight, plasma lipids and liver or visceral adipose tissue gene expression are observed, atherosclerotic lesion size does not change, in either model. The authors conclude that ClnA does not have an impact on atherosclerotic plaque development in this model.

Comments:

·    The treatment may have been not long enough, even if given for 18 weeks. This should be discussed.

·     Also, in the SHORT protocol, the treatment was started at age of 18 weeks, and atherosclerosis lesions have already started to develop. Therefore, the wording atherosclerosis development should be reconsidered and changed to atherosclerosis progression.

·    Use of apoE-/- LDLR-/-: explain why this selection and not only apoE or LDLR deficient mice. The combination of both is a very strong stimulus, and a nutritional ingredient may not potent enough to overturn its effects.

·  The information on absent changes in cellular composition of atherosclerotic lesions should be added to the abstract.

·   Please indicate whether male and female mice were examined and if both, if data were pooled. Sex is known to have effects on atherosclerosis.

·    Was a statistical power calculation of the number of mice needed to find an expected difference performed a priori? The number (and sex) of mice per group needs to added to Table 1 and 2.

·  Only CD68 was used to study plaque inflammation, but there are additional markers that also distinguish between pro- and anti-inflammatory macrophage subtypes. As no differences in lesion size are found, more effort should go into the analysis of cellular lesion composition. Especially, as the manuscript is submitted for a Special Issue on Inflammatory Response in Endocrine Disorders.

· Figures 1-4: give the number of mice examined and explain the meaning of a, ab etc.

·  The Discussion is very detailled and could be shortened and more focussed. Also, a separate section discussing the limitations of the study should be included at the end.

·   Abstract: please specificy in which cell or organ the changes in gene expression were observed. Also explain the abbreviation CLA.

Author Response

Response to Reviewer 1 Comments:

  •   The treatment may have been not long enough, even if given for 18 weeks. This should be discussed.

Thank you very much for the Reviewer’s comment. We are aware that the experimental duration is very important. However, atherosclerotic plaque development in ApoE/LDLR-/- mice model has been described by our group previously (Csanyi et al., 2012). We observed that atherosclerotic plaque in ApoE/LDLR-/- develops much faster comparing to single knockout. Therefore, based on our previous studies we decided that the experimental design will be divided to two parts (Short – regression and Long - progression).

Csányi,G.; Gajda, M.; Franczyk-Zarow, M.; Kostogrys, R.; Gwoźdź, P.; Mateuszuk, Ł.; Sternak, M.; Wojcik, L.; Zalewska, T.; Walski, M.; Chlopicki, S. Functional alterations in endothelial NO, PGI₂ and EDHF pathways in aorta in ApoE/LDLR-/- mice. Prostaglandins Other Lipid Mediat. 2012, 98(3-4):107-15.

  •    Also, in the SHORT protocol, the treatment was started at age of 18 weeks, and atherosclerosis lesions have already started to develop. Therefore, the wording atherosclerosis development should be reconsidered and changed to atherosclerosis progression.

Thank you for the comment. However, in the Short protocol we assumed that due to existing atherosclerosis lesions the potential regression of the changes will be assessed.

  •   Use of apoE-/- LDLR-/-: explain why this selection and not only apoE or LDLR deficient mice. The combination of both is a very strong stimulus, and a nutritional ingredient may not potent enough to overturn its effects.

We agree with Reviewer that the double knockout mice model (ApoE/LDLR-/-) is very strong stimulus. However, as shown in Table 2, concentration of TCh and LDL were significantly decreased after CLnA diet in the Short experiment. We know that in the single knockout mice models (ApoE or LDLR deficient mice), the high fat diet should be used. Such a factor could interfere the results of study. Using only the double knockout mice model (ApoE/LDLR-/-) it is possible to avoid the diet modification (such as high fat diet). DKO (ApoE/LDLR-/-) mice develop the atherosclerosis spontaneously.

  • The information on absent changes in cellular composition of atherosclerotic lesions should be added to the abstract.

Thank you very much for the Reviewer comment. However, we didn’t assess the atherosclerotic plaque composition.

  •  Please indicate whether male and female mice were examined and if both, if data were pooled. Sex is known to have effects on atherosclerosis.

In the study the female mice were used.

  •   Was a statistical power calculation of the number of mice needed to find an expected difference performed a priori? The number (and sex) of mice per group needs to added to Table 1 and 2.

Thank you very much for the Reviewer comment. The number of mice per group (n=10) has been added to Table 2 and 3. The statistical power calculation has not been used in the study.

  • Only CD68 was used to study plaque inflammation, but there are additional markers that also distinguish between pro- and anti-inflammatory macrophage subtypes. As no differences in lesion size are found, more effort should go into the analysis of cellular lesion composition. Especially, as the manuscript is submitted for a Special Issue on Inflammatory Response in Endocrine Disorders.

Thank you very much for the Reviewer’s comment. We are aware that the plaque composition is very important. However, the immunohistochemical analysis of tissues has not been performed due to financial limitations. 

  • Figures 1-4: give the number of mice examined and explain the meaning of a, ab etc.

The number of mice (n=10) has been added to Figures 2-6. The means marked with letters a and b are significantly different (p<0.05).

  • The Discussion is very detailled and could be shortened and more focussed. Also, a separate section discussing the limitations of the study should be included at the end.

According to the Reviewer’s comment, we shortened the Discussion section. Additionally we included the limitation paragraph at the end of Discussion section.

  • Abstract: please specificy in which cell or organ the changes in gene expression were observed. Also explain the abbreviation CLA.

According to the Reviewer’s suggestion in the Abstract the abbreviation of CLA has been explained as well as we specified the organ (i.e. liver) where gene’s expression changes were observed.

Reviewer 2 Report

This is a very nice work dealing with studying and assessment of antagonistic effect to atherosclerosis attributed to CLnA in the pomegranate seed oil. The comparison to LnA and CLA was done in the mice model system.

The outcomes declare that CLnA had no significant or any influence whatsoever on the atherosclerosis development and progression. However, it has been reported that lipid profile and metabolism can be affected.

The length of manuscript and concept are fairly acceptable and clearly reported confirmed by exhaustive experimental part and results.

For me, manuscript is acceptable at current form

Author Response

Response to Reviewer 2 Comments:

This is a very nice work dealing with studying and assessment of antagonistic effect to atherosclerosis attributed to CLnA in the pomegranate seed oil. The comparison to LnA and CLA was done in the mice model system.

The outcomes declare that CLnA had no significant or any influence whatsoever on the atherosclerosis development and progression. However, it has been reported that lipid profile and metabolism can be affected.

The length of manuscript and concept are fairly acceptable and clearly reported confirmed by exhaustive experimental part and results.

For me, manuscript is acceptable at current form

Thank you very much for the opinion of the Reviewer 2.

Reviewer 3 Report

This study investigated the anti-atherosclerotic effect of pomegranate seed oil as a source of CLnA in apoE/LDLR-/- mice. The effects of the CLnA diet on physiological indices, plasma lipid profile, and expression of PPARα and SREBP-1 as well as their target genes in mice were determined by short-term and long-term experiments and found that CLnA improved lipid profile and affected the expression of lipid metabolism genes. But no effect was found on the development of atherosclerotic plaques in apoE/LDLR-/- mice. Overall, these results are positive for understanding the bioactive function of CLnA. I recommend a minor revision before its acceptance.

1. The data should have no more than three significant digits.

2. Page 2, Line 83-84: This sentence seems abrupt, and I suggest the authors to move it to “Section 4.2. Diets and feeding”.

3. Please check the entire text for spelling errors and check grammar. For example, on Page 5, Line 498.

4. What was the basis for the quantification of “Diets were supplied with seed oils” in Table 1? What is the quantitative-effect relationship between oil supplementation and health effects?

5. Page 9, Line 73, The experimental results corresponding to the "Real-time qRT-PCR" section seem to be missing, and it is suggested to add the data of this section as a separate part of the results.

6. The cited literature is too old, the references of the last five years are quite a few, and it is recommended to cite the latest literature.

Author Response

Response to Reviewer 3 Comments:

Point 1. The data should have no more than three significant digits.

As required the data in Table 1 has been changed to g/kg, so there is no more than three significant digits.

Point 2. Page 2, Line 83-84: This sentence seems abrupt, and I suggest the authors to move it to “Section 4.2. Diets and feeding”.

According to Reviewer’s suggestion the sentence has been moved to “Section 4.2. Diets and feeding”.

Point 3. Please check the entire text for spelling errors and check grammar. For example, on Page 5, Line 498.

We apologize for spelling mistakes. Two errors with “then” instead of “than” has been corrected.

Point 4. What was the basis for the quantification of “Diets were supplied with seed oils” in Table 1? What is the quantitative-effect relationship between oil supplementation and health effects?

The oil content of each experimental diet was balanced at 70 g/kg diet based on the AIN93G formulation (Reeves et al. 1993). The detailed composition is shown in Table 1. Our goal was that “diets were supplied with seed oils equivalent to an amount of 0.5% of studied fatty acids”. In our work, we studied the effect of a conjugated fatty acid (trien CLnA), its potential precursor form (non-conjugated LnA) and the form in which the triene is metabolized in the body - the conjugated isomer of linoleic acid (diene CLA).

So after calculating the exact composition (Flaxseed oil contained 48.8 g LnA/100 g, Pomegranate seed oil contained 57.5 g CLnA/100 g and CLA isomers cis-9,trans-11 CLA methyl ester, purity 75%) it was decided that 10.25 g of flaxseed oil, 8.7 g of pomegranate seed oil and 6.66 g of CLA-isomer would be added to the diets.

Point 5. Page 9, Line 73, The experimental results corresponding to the "Real-time qRT-PCR" section seem to be missing, and it is suggested to add the data of this section as a separate part of the results.

 According to Reviewer’s suggestion the data of the "Real-time qRT-PCR" section has been presented as a separate part of results in Table 3. The numbering of the tables in the manuscript has been corrected.

Point 6. The cited literature is too old, the references of the last five years are quite a few, and it is recommended to cite the latest literature.

Thank you for the Reviewer’s recommendation. Recent literature on the topic has been added [e.g. 32, 40].

Round 2

Reviewer 1 Report

The authors only partly responded to my comments, in particular comments 1-3 and 5. Especially as they report negative findings, all possibilities for this need to be discussed in the manuscript, not only with me. In the light of metabolic changes, it is surprising to find no changes in atherosclerotic plaque size. There are several potential reasons for this based on the study design and the few readouts (only oil red o). They have to be discussed.

Author Response

Response to Reviewer 1 Comments:

The authors only partly responded to my comments, in particular comments 1-3 and 5. Especially as they report negative findings, all possibilities for this need to be discussed in the manuscript, not only with me. In the light of metabolic changes, it is surprising to find no changes in atherosclerotic plaque size. There are several potential reasons for this based on the study design and the few readouts (only oil red o). They have to be discussed.

Thank you very much for the Reviewer comment. We improved the manuscript. According to the Reviewer’s suggestion we added the missing information in the limitations paragraph as well as in the Materials and methods section. We also discussed the lack of the changes of the atherosclerotic plaque area despite the decreased values in TCh and LDL cholesterol in the SHORT study. Moreover, additional references have been added.
